
# Evaluating the vegetation-atmosphere coupling strength of ORCHIDEE land surface model (v7266)

Yuan Zhang[1,2], Devaraju Narayanappa[1], Philippe Ciais[1], Wei Li[3], Daniel Goll[1], Nicolas Vuichard[1], Martin G. De Kauwe[4], Laurent Li[2], Fabienne Maignan[1]

[1]*Laboratoire des Sciences du Climat et de l'Environnement (LSCE), IPSL, CEA/CNRS/UVSQ, Gif sur Yvette, France*
[2]*Laboratoire de Météorologie Dynamique, IPSL, Sorbonne Université/CNRS, Paris, France*
[3]*Department of Earth System Science, Ministry of Education Key Laboratory for Earth System Modeling, Institute for Global Change Studies, Tsinghua University, Beijing, 100084, China*
[4]*School of Biological Sciences, University of Bristol, Bristol, BS8 1TQ, UK*

*Correspondence: Yuan Zhang (yuan.zhang@lsce.ipsl.fr)*

**Abstract.** Plant transpiration dominates terrestrial latent heat fluxes (LE) and plays a central role in regulating the water cycle and land surface energy budget. However, currently Earth system models (ESM) disagree strongly on the amount of transpiration, and thus LE, leading to large uncertainties in simulating future climate. Thus it is crucial to correctly represent the mechanisms controlling the transpiration in models. At the leaf-scale, transpiration is controlled by stomatal regulation, and at the canopy-scale, through turbulence, which is a function of canopy structure and wind. The coupling of vegetation to the atmosphere can be characterized by a coefficient $\Omega$. A value of $\Omega \rightarrow 0$ implies a strong coupling of vegetation and the atmosphere, leaving a dominant role to stomatal conductance in regulating water ($H_2O$) and carbon dioxide ($CO_2$) fluxes, while $\Omega \rightarrow 1$ implies a complete decoupling of leaves from the atmosphere, that is, the transfer of $H_2O$ and $CO_2$ is limited by aerodynamic transport. In this study, we investigated how well the land surface model ORCHIDEE (v7266), simulates the coupling of vegetation to the atmosphere by using empirical daily estimates of $\Omega$ derived from flux measurements from 106 FLUXNET sites. Our results show that ORCHIDEE generally captures the $\Omega$ in forest vegetation types (0.27±0.10) compared with observation (0.26±0.09), but underestimates $\Omega$ in grasslands and croplands (0.26±0.16 for model, 0.33±0.17 for observation). The good model performance in forests is due to compensation of biases in surface conductance (Gs) and aerodynamic conductance (Ga). Calibration of key parameters controlling the dependence of the stomatal conductance to the water vapor deficit (VPD) improves the simulated Gs, and $\Omega$ estimates in grasslands and croplands (0.30±0.21). To assess the underlying controls of $\Omega$, we applied random forest (RF) models to both simulated and observation-based $\Omega$. We found that large observed $\Omega$ are associated with periods of low wind speed, high temperature, low VPD and related to sites with large leaf area index (LAI) and/or short vegetation. The RF models applied to ORCHIDEE output generally agree with this pattern. However, we found the ORCHIDEE underestimated the sensitivity of $\Omega$ to VPD when VPD is high, overestimated the impact of LAI on $\Omega$, and did not correctly simulate the temperature dependence of $\Omega$ when temperature is high. Our results highlight





the importance of observational constraints on simulating the vegetation-atmosphere coupling strength, which can help improve predictive accuracy of water fluxes in Earth system models.

## 1 Introduction

Representing accurately the land-atmosphere interactions in Earth system models (ESMs) is crucial for analyzing climate variability and projecting climate change (Claussen, 1998; Goldberg and Bernhofer, 2001; Zhu et al., 2017). Among the key interactions, the exchange of latent heat (LE) between the land surface and the atmosphere is one of the most important processes (Trenberth et al., 2009). LE is contributed by several sources, including evaporation from bare soil and canopy interception, vegetation transpiration, snow and ice sublimation. In these sources, transpiration has the largest contribution

(Jasechko et al., 2013), but is massively uncertain across models (Stoy et al., 2019), leading to considerable uncertainty in LE simulation in current ESMs (Wild, 2020). The large uncertainties in current transpiration and LE simulations can further result in difficulties in constraining soil moisture and the carbon cycle (Humphreys et al., 2021). Therefore, there is a need to evaluate and improve the simulation transpiration and LE in ESMs.

The LE parameterization in ESMs is based on Fick's law, using the conductance, or 1/resistance of water vapor between

vegetation and atmosphere (Bonan, 2019). This conductance is the sum of several processes such as stomatal opening, boundary layer turbulence, soil-to-air evaporative resistance, and it is thus affected by multiple factors including plant physiology, vegetation structure, vapor pressure deficit (VPD), temperature, net radiation, soil moisture etc. Currently, we can observe total LE at the site scale (i.e. FLUXNET), but we are unable to disentangle the relative contribution of different processes. The complexity of conductance and the lack of process-level observations lead to difficulties in detailed evaluation

on the vegetation-atmosphere water exchanges in ESMs based on the underlying processes. As a result, accurately capturing the regulation of LE by biotic and abiotic factors remains a key challenge for the land surface modeling community (De Kauwe et al., 2017).

An early attempt to quantify the contribution of different conductance processes was made by Jarvis and McNaughton (1986), who developed a metric commonly referred to as the decoupling coefficient, $\Omega$, to describe whether vegetation transpiration

is mainly controlled by stomatal or aerodynamic processes. The calculation of $\Omega$ is based on the ratio between aerodynamic and stomatal conductance (See Method). At the limit, $\Omega=0$ denotes perfect coupling between vegetation and atmosphere, i.e. the transpiration is entirely regulated by stomata, while $\Omega=1$ denotes complete decoupling, i.e. transpiration is driven entirely by boundary layer turbulence. The concept of $\Omega$ can be used at scales from leaf to regional level, and for different fluxes from transpiration only to the total evapotranspiration, for instance, Peng et al. (2019). Because evapotranspiration includes water

fluxes from not only leaf but also other surfaces, the stomatal conductance needs to be replaced by a surface conductance which sums all conductances at different surfaces in the evapotranspiration $\Omega$ calculation.

During the last decades, the number of eddy covariance flux measurements has rapidly grown. Quantification of $\Omega$ at site level from eddy covariance flux measurements offers insights into how different vegetation types control turbulent fluxes as a



function of their phenology and stomatal physiology during the growing and the non-growing season (De Kauwe et al., 2017; Goldberg and Bernhofer, 2001). These observation-based $\Omega$ provides valuable information to evaluate ESMs on how well they capture the controls of LE. Using this estimates, De Kauwe et al. (2013) found that one of the principal reasons for disagreement among simulated transpiration responses to elevated $CO_2$ is the differences in the degree of coupling between vegetation and the atmosphere.

ORCHIDEE land surface model (LSM) is one of the widely used models in simulating carbon, energy and water budget of terrestrial ecosystems (e.g. Zhang et al., 2021; Schrapffer et al., 2020). ORCHIDEE and the ESM IPSLCM, which has ORCHIDEE as the land surface module have participated in various model intercomparison projects including TRENDY, Coupled Model Intercomparison Project (CMIP), etc. In spite of its wide usage, the LE of ORCHIDEE LSM remains simply calibrated and evaluated against the total evapotranspiration observations (Bastrikov et al., 2018), without considering the detailed processes. A recent study showed that the ORCHIDEE version used in CMIP6 still has biases in LE, especially in tropical regions (Tafasca et al., 2020). However, it remains unclear how the biases happened and which processes need to be improved to better simulate the fluxes. To solve this problem, in this study we used $\Omega$ dataset derived from eddy-covariance data from 106 sites (De Kauwe et al., 2017), to evaluate the vegetation-atmosphere coupling strength of the land surface model ORCHIDEE 2.2 (v7266). We tested whether the calibration of the stomatal response to atmospheric dryness, or using observed canopy height, can improve the simulation of coupling strength. Further we used random forest models to investigating the biotic and abiotic factors affecting the decoupling strength. The methodology presented here is generic enough to be applied for the benchmarking of other LSMs. The objectives of this study are to: (1) Benchmark ORCHIDEE using $\Omega$ estimated from FLUXNET observations; (2) Investigate how different factors affect $\Omega$ in the observations and whether ORCHIDEE correctly captured the driving factors.

## 2 Data and methods

### 2.1 ORCHIDEE model

We use the ORCHIDEE 2.2 (v7266) land surface model in this study. This model version is the latest version participating in CMIP6 project under coupled configuration to atmospheric circulation model in the IPSL-CM6A-LR ESM (Boucher et al., 2020). The ORCHIDEE model consists of three interactive sub-modules (Krinner et al., 2005). The SECHIBA module parameterizes the land surface energy and water balance (Ducoudr éet al., 1993). The STOMATE module deals with phenology (Botta et al., 2000) and carbon fluxes of terrestrial ecosystems (Viovy, 1996). The LPJ dynamic vegetation module simulates the dynamics of vegetation (Sitch et al., 2003). In this study, the dynamic vegetation module is turned off because the vegetation types are prescribed at each site.

ORCHIDEE simulates LE by considering plant transpiration, bare soil evaporation, sublimation, floodplain evaporation, and evaporation from canopy water interception. Because this study focuses on the vegetation-atmosphere decoupling strength for


transpiration and also because the data to evaluate this model has been filtered to represent the transpiration (De Kauwe et al., 2017), here we only introduce the parameterization of conductance relating to transpiration in ORCHIDEE.

The stomatal conductance (gs, mol m$^{-2}$ s$^{-1}$ bar$^{-1}$) is calculated in the photosynthesis module which couples the leaf-level photosynthesis and stomatal conductance based on (Yin and Struik, 2009):

$$gs = g0 + \frac{A+R_d}{C_i-C_i^*} f_{vpd} \tag{1}$$

Where g0 is the stomatal conductance when the irradiance is zero (mol m$^{-2}$ s$^{-1}$ bar$^{-1}$). A is the rate of $CO_2$ assimilation (μmol m$^{-2}$ s$^{-1}$), $R_d$ is the dark respiration (μmol m$^{-2}$ s$^{-1}$), $C_i$ is the intercellular $CO_2$ partial pressure (μbar), $C_i^*$ is the $C_i$-based $CO_2$ compensation point (μbar) in the absence of Rd, and $f_{vpd}$ is the function for the effect of vapor pressure deficit (VPD, kPa) on stomatal conductance, calculated as:

$$f_{vpd} = \frac{1}{[\frac{1}{a_1-b_1 VPD}-1]} \tag{2}$$

Here $a_1$ and $b_1$ are empirical parameters depending on vegetation type (Fig S1). This equation shows that a higher VPD will induce stomatal closure and decrease gs.

The aerodynamic conductance ($G_a, mol\ m^{-2}\ s^{-1}$) formulation in ORCHIDEE is

$$G_a = \frac{\kappa^2 u_a}{\left[ ln\left(\frac{z-d}{z_{0m}}\right) ln\left(\frac{z-d}{z_{0h}}\right) \right]} \tag{3}$$

where $z$ is the average height of all PFTs (including bare soil) in a grid (m), d is the displacement height (i.e. the height at

which the wind speed would go to zero), calculated as 0.66 of z. $u_a$ is wind speed ($ms^{-1}$), $\kappa$ is the von Karman's constant, $z_{0m}$ and $z_{0h}$ are respectively the roughness heights (m) for momentum and heat transfer estimated following Su et al. (2001) and Ershadi et al. (2015) using canopy height and LAI:

$$z_{0m} = (z-d)e^{-\frac{\kappa}{\eta}} \tag{4}$$

Where

$$\eta = 0.32 - 0.264e^{-3.02LAI} \tag{5}$$

$z_{0h}$ is estimated using $z_{0m}$ (see Eq E2 in Ershadi et al. (2015)).

**2.2 FLUXNET data and simulation setup**

The site simulations with ORCHIDEE are forced with observed meteorology in the FLUXNET 2015 dataset (Pastorello et al., 2020). The variables include half-hourly time series of air temperature (K), surface pressure (Pa), specific humidity (kg kg$^{-1}$),

North and East direction wind speed (m s$^{-1}$), short-wave down (W m$^{-2}$), long-wave down (W m$^{-2}$), rainfall (kg m$^{-2}$ s$^{-1}$) and snowfall (kg m$^{-2}$ s$^{-1}$). Gaps in the FLUXNET meteorology data are filled following Vuichard and Papale (2015). The plant functional type (PFT) classification of FLUXNET is different from the one used in ORCHIDEE. To let ORCHIDEE simulate LE and the conductances without bias, we used a combination of ORCHIDEE PFT types to represent the vegetation type at each site. The detailed information of flux sites can be found in Table S1.





Three simulations are performed at each site (Fig. 1). The first simulation named *Ctrl* uses the default configuration and parameters as used in CMIP6 and TRENDY experiments. The second simulation named *Clb_gs* uses the same configuration as *Ctrl* but changes the empirical parameters in Eq. 2. New values for $a_1$ and $b_1$ are obtained by constraining the modeled formulation of conductance against a global database of leaf-level observations of stomatal conductance from Lin et al. (2015) for different plant functional types (See the Supplementary, Table S2, Fig S1). Finally, because the ORCHIDEE model

prescribes canopy height for each PFT (Table S3), which may cause biases in Ga, we performed a last simulation referred to as *Clb_ht*. *Clb_ht* also uses the *Ctrl* configuration but the default canopy height parameters for each PFT are replaced by the canopy height observed at each site. Because canopy height is required in the last simulation, we only used 106 sites where we found height information out of the flux sites in the FLUXNET2015 dataset in this study.

### 2.3 Empirical calculation of Ω

The calculation of Ω was firstly introduced by Jarvis and McNaughton (1986), using the formulation:

$$\Omega = \frac{1+\epsilon}{1+\epsilon+\frac{G_a}{G_s}} \tag{6}$$

where $\epsilon = \frac{s}{\gamma}$, $s$ is the slope of the saturation vapor pressure curve with air temperature (Pa K$^{-1}$), $\gamma$ is the psychrometric constant (Pa K$^{-1}$). It should be noted that the conductance ($Ga$, $Gs$) used for Ω calculation depends on the scale of interest, at the scale larger than a leaf, if other water vapor fluxes besides transpiration (e.g. soil evaporation) have significant contribution to LE,

$Gs$ must also include such contribution. In such cases, the synthesized $Gs$ was sometimes referred to as surface conductance (Peng et al., 2019). To be accurate, we use the term surface conductance for $Gs$ hereafter to match our scale.

De Kauwe et al. (2017) derived an Ω dataset over the sites of the FLUXNET network. In their calculation, Ga was estimated as an empirical equation using wind speed and friction velocity (Thom et al., 1975), and Gs ($mol\ m^{-2}\ s^{-1}$) was estimated using inverted Penman–Monteith equation with measured evapotranspiration (*ET*, in mol m$^{-2}$ s$^{-1}$) flux:

$$G_s = \frac{G_a \gamma \lambda ET}{s(R_n - G) - (s+\gamma)\lambda ET + G_a M_a cVPD} \tag{7}$$

Where $\lambda$ is the latent heat of vaporization (J mol$^{-1}$), *VPD* (Pa) is the vapor pressure deficit, $R_n$ (Wm$^{-2}$) is the net radiation flux, $G$ (W m$^{-2}$) is the soil heat flux, $M_a$ (kg mol$^{-1}$) is molar mass of air, and $c$ is the heat capacity of air (J kg$^{-1}$ K$^{-1}$).

Although De Kauwe et al. (2017) excluded time steps with precipitation and the subsequent 48 half hours to have the LE mainly contributed by transpiration and referred to Gs as 'stomatal conductance' in their paper, we still need to keep in mind

that the Gs calculated in this way may also contain contributions from several other processes. It includes the conductance related to bare soil evaporation and the one related to water transport in the leaf boundary layer, in addition to the stomatal conductance integrated over the entire canopy. So it is more a 'surface conductance' than a 'stomatal conductance". To be consistent with the observation-based dataset, we did not use the integrated canopy level stomatal conductance from ORCHIDEE output to calculate Ω. Instead, *Gs* is diagnosed using ORCHIDEE output *ET*, $R_n$ and $G$ following Eq 7.





**2.4 Leaf area index data**

Because leaf area is an important factor affecting both aerodynamic and surface conductance, it is necessary to take leaf area into consideration when explaining the decoupling coefficient. However, instantaneous leaf area information is not available at most of the flux site. To match the space and time of observation-based $\Omega$, we extracted the leaf area index (LAI) from the 500m 8-day MOD15A2H dataset derived from the space-borne MODIS observations (Myneni et al., 2015). This LAI dataset

shows good consistency with in situ observations (Xu et al., 2018). The LAI for a given date is interpolated by averaging the nearest two high-quality LAI observations from the 8-day time series. For the simulated $\Omega$, we used the LAI from the simulations for analyses to keep consistency between $\Omega$ and LAI.

**2.5 Analyses**

To be comparable with the observation-based $\Omega$ dataset, we first used the same criteria to screen the model outputs as De

Kauwe et al. (2017), i.e. (1) only the three most productive months, to account for the different timing of summer in the Northern (June, July, August) and Southern (December, January, February) hemispheres are included in the study. This is to maximize the role of transpiration in $\Omega$ versus bare soil evaporation in the growing season. (2) only day-time data from 8:00 am to 16:00 pm (local solar time) are used. (3) time steps during precipitation or within 2 days after precipitation are excluded. Because the 30-min $\Omega$ is very noisy, to reduce the noise in data, we used the day-time average of $\Omega$ and explanatory variables

in all later analyses.

The decoupling strength $\Omega$ is affected by multiple factors and the relationships between $\Omega$ and different factors are often nonlinear. To characterize these relationships, we constructed random forest models for each of the observation-/simulation-based daily $\Omega$. The goal is here to diagnose the main explanatory variables from the random forests in the observations/simulations, and to gain insights about the model over-/under-representation of their relative importance. The

explanatory variables used in the random forest models include wind speed, air temperature (Tair), VPD, net radiation (Rnet), LAI, canopy height and PFT. For each model, 90% of the data are randomly sampled for training and the left 10% are used for testing whether there is overfitting in the random forest models (Fig S2).

To visualize the role of each factor in the complex random forest model, we calculated the SHapley Additive exPlanations (SHAP) values. SHAP value is an index based on the classic Shapley values from game theory (Lundberg and Lee, 2017). For

each daily sample, SHAP calculates the expectation of contribution of each factor to deviate the sample value from the average of all samples. An example explaining the SHAP values can be found in Fig S3. Investigating the dependence of SHAP value to the factor value tells how this factor affects $\Omega$. Also, by averaging the absolute values of the SHAP of one factor from all samples, we can get the importance of the factor in the random forest model.

The workflow of the simulations and analyses can be found in Fig 1.




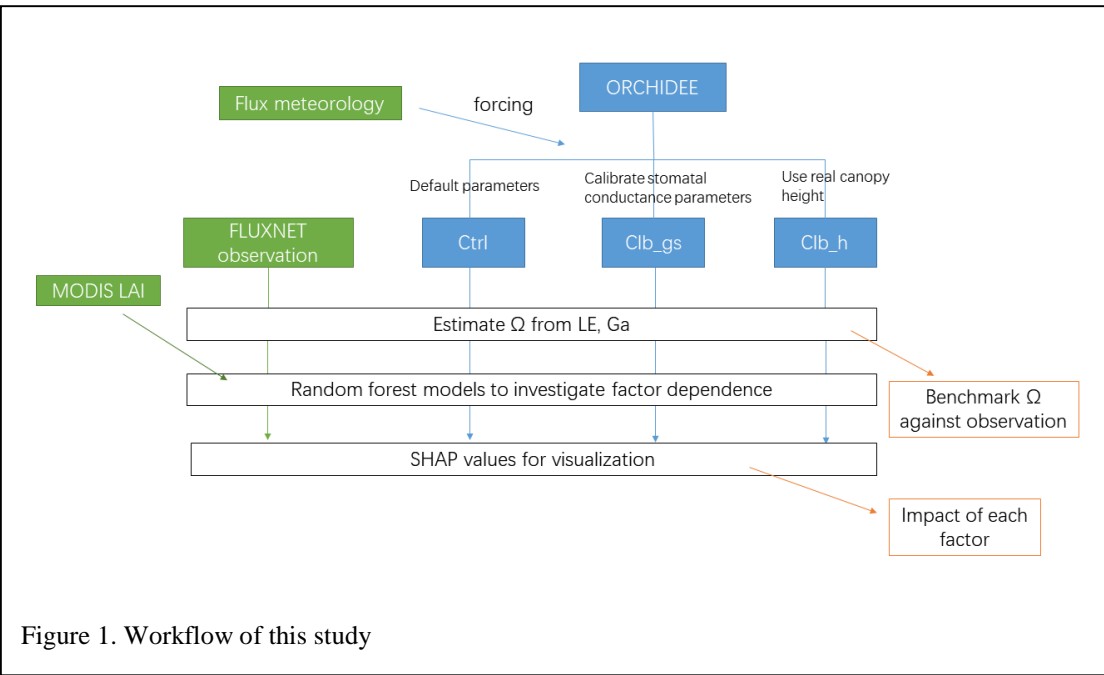

Figure 1. Workflow of this study

## 3 Results

### 3.1 The performance of the ORCHIDEE model

The average growing season daytime Ω estimated from observations and from the ORCHIDEE outputs are shown in Fig 2. A remarkable difference in the decoupling coefficient is found among plant functional types. According to the observation-based estimation (De Kauwe et al. 2017), the short vegetation types including grasslands (GRA) and croplands (CRO) are generally more decoupled from the atmosphere than forests, with the median values of Ω over sites of 0.31 and 0.38. In forest vegetation types, the broadleaf forests (median Ω=0.29-0.33) are more coupled with the atmosphere than needleleaf forests (median Ω=0.22). The wetlands in observation show a strong decoupling (median Ω=0.46). Considering the large evaporation from open water in this vegetation type, the strong decoupling is not surprising. Besides the difference among vegetation types, we also find large variability in Ω within each type, especially for GRA and CRO (Table S4).

Compared with observations, ORCHIDEE *Ctrl* simulations show similar median Ω in forests and croplands (Fig 2, Table S4). However, in grasslands, the *Ctrl* median Ω (0.15) is much smaller compared to observation (0.31), implying a greater stomatal control in the model than the observations on grassland transpiration. This bias is not contributed by a few outlier sites but by a systematic underestimation of Ω at most of the grassland sites (Fig S4). For wetlands, ORCHIDEE also shows a significant underestimation of Ω (Fig 2). This could be due to the lack of wetland PFT and the corresponding open water in the ORCHIDEE model (Table S3). In spite of the biases in grassland and wetland, the observed differences in Ω among vegetation



type are to a larger degree well reproduced (Fig 2). The strongest decoupling is found in CRO and deciduous broadleaf forest (DBF), and the needleleaf forests are more coupled than deciduous broadleaf forests.

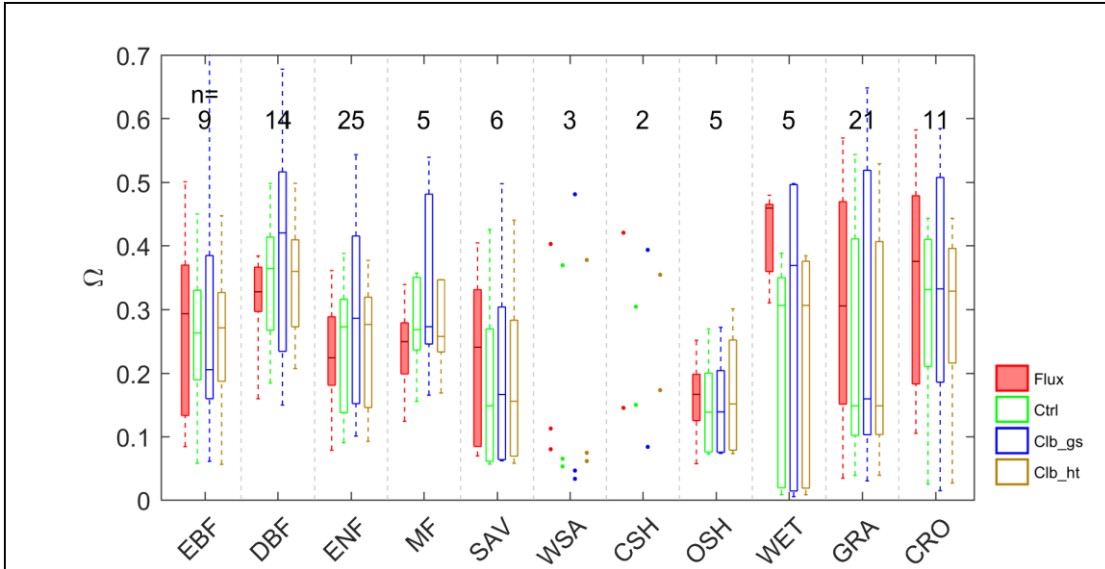

Figure 2. Box plots of site mean Ω observation (Flux) and different simulations, n indicates the number of sites in each PFT group. EBF, Evergreen broadleaf forests; DBF, Deciduous broadleaf forests; ENF, Evergreen needleleaf forests; MF, Mixed forests; SAV, Savannas; WSA, Woody savannas; CSH, Closed shrublands; OSH, Open shrublands; WET, wetlands; GRA, grasslands; CRO, croplands

By calibrating stomatal conductance (VPD dependence parameters leading to the *Clb_gs* simulation), we obtained Ω estimations closer to observations in short vegetation types (CRO and GRA) than *Ctrl* (Fig 2). But the median Ω estimation for most forest types is degraded after the gs 'calibration', with the Ω more overestimated in DBF, ENF and MF, but more underestimated in EBF. In contrast to the large impact from the calibration of stomatal conductance, prescribing realistic canopy height to the model leads to minor changes in Ω (Fig 2).

In order to understand the reasons for differences in Ω between observation and the ORCHIDEE model, we also look into its components Ga and Gs (Fig 3). Compared to observation, both Ga and Gs are underestimated in *Ctrl*. For Ga, the underestimation from model is 0.5-0.8 mol $m^{-2}$ $s^{-1}$ in forest types and 0.2-0.3 mol $m^{-2}$ $s^{-1}$ in GRA and CRO. Calibrating stomatal conductance (*Clb_gs*) or prescribing the observed canopy height to the model (*Clb_ht*) both have a small impact on Ga. For Gs, using the new parameters for stomatal conductance (*Clb_gs*) can generally correct the Gs bias in EBF, DBF and ENF, and improved Gs in GRA and CRO than *Ctrl*. Although *Clb_gs* has improved the Gs simulation compared with *Ctrl*, it does not result in an improvement of Ω and latent heat simulation, implying a compensate of biases in Ga and Gs in current ORCHIDEE model.



Figure 3. Same as Fig. 2 but for (a) aerodynamic conductance, (b) surface conductance and (c) latent heat.

## 3.2 Factors controlling the decoupling strength





To better understand the underlying drivers of the variability in decoupling we separated the importance of hypothesized drivers of decoupling strength in random forest models using SHAP values (Fig. 4a). Among all the factors, the observation-based random forest results show that the variation of $\Omega$ is mainly contributed by the variation of VPD, followed by PFT, with

each of them having a SHAP value of ~0.06, i.e. the variation of the factor contributes on average 0.06 of the deviation of $\Omega$ (absolute value) from the average of all samples. The other factors show relatively small importance to $\Omega$, with SHAP values smaller than 0.03. Compared to observations, the ORCHIDEE $\Omega$ variation is also strongly contributed by VPD. However, opposite from the strong PFT impact found in observation, the modeled $\Omega$ is strongly affected by LAI. In *Ctrl*, the SHAP value of LAI is 0.09, which is much higher than the observation. The calibration of gs increased this value to 0.13. In contrast to the

strong impact of LAI, all the modeled $\Omega$ show a much smaller contribution from PFT than in observation. It is also notable that the impact of air temperature on $\Omega$ is also much smaller in ORCHIDEE simulations than in observations.

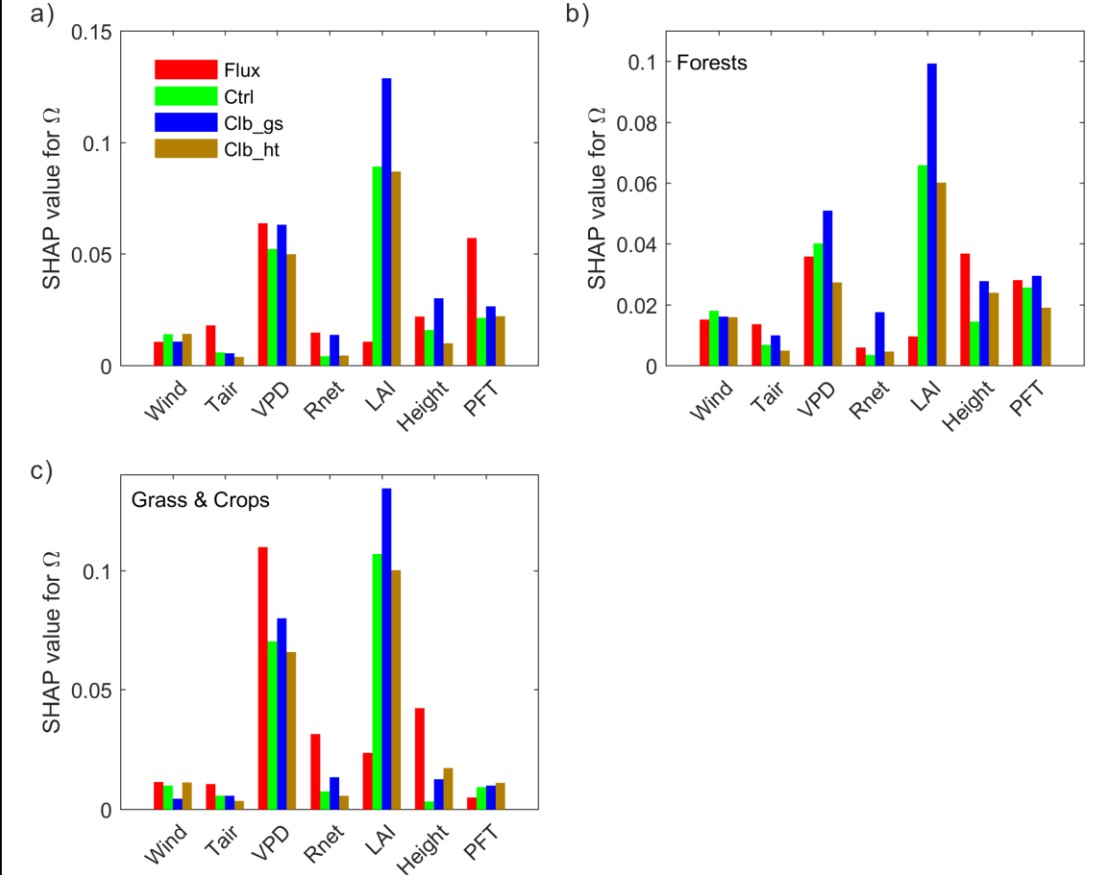

Figure 4. Importance of different factors based on absolute SHAP values of $\Omega$ (the expectance of factor-induced deviation of $\Omega$ from the averages of all samples). (a) in random forest model built by data from all PFTs. (b) model using only forest data. (c) model using only grassland and cropland data.

To further understand the differences between tall and short vegetation, we trained random forest models using only forests (EBF, DBF, ENF and MF) and only short vegetation (GRA and CRO) observation/simulation. In forests, the SHAP value of VPD is comparable in the observation and ORCHIDEE simulations, while the LAI SHAP value is strongly overestimated and

the canopy height SHAP value is slightly underestimated by the model. For short vegetation, a strong overestimation of the SHAP of LAI is also confirmed in ORCHIDEE. But for the other factors (Tair, Rnet, VPD and height), the SHAP values are underestimated. It is notable that the SHAP values for VPD in ORCHIDEE is only 60% of the estimation in observation, probably indicating a strong underestimation of water stress on Ω in short vegetation.

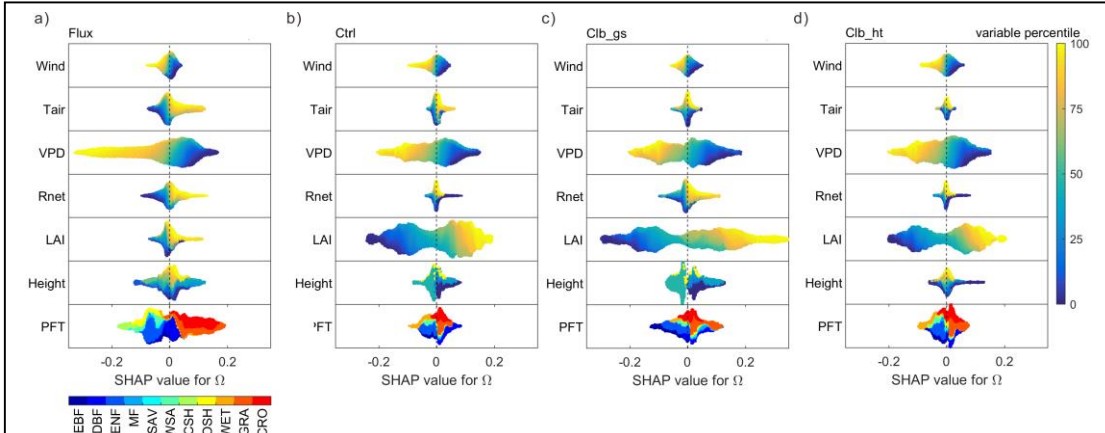

Figure 5. Beeswarm plots showing the dependence of Ω SHAP values to different factors. For each data point, the percentile of the factor's value in all samples is shown in color. The SHAP value, or contribution of this factor value to deviate the day-time Ω from the average Ω of all samples, is shown in x axis. In each subplot, data points at a certain SHAP value level are sorted by the factor percentile (i.e. vertical gradient indicates the distribution of factor values in the data). (a) based on observation dataset, (b), (c) and (d) are for Ctrl, Clb_gs and Clb_ht simulations respectively.

Figure 5 summarizes how different factors affect Ω in each of the observation/simulation random forest models. The responses

of Ω to most factors are generally consistent in observations and simulations. According to all of the random forest models, the vegetation is more decoupled, or having a larger Ω, under conditions with low wind speed, low VPD and large LAI. Also, both observation and simulations agree that GRA and CRO are more decoupled from the atmosphere than the other PFTs. However, for Tair and Rnet, ORCHIDEE does not capture the observed dependence correctly. In observation, a remarkable positive Tair dependence is found, with higher temperature tending to result in higher Ω. While in simulations, temperature

shows a very small impact on Ω. Furthermore, for *Clb_gs* and *Clb_ht* simulations, the low Tair tends to result in large Ω. The dependence of Ω on Rnet is similar to that of Tair in observation, but only the *Clb_gs* simulation captured this dependence correctly. Finally, to our surprise, we did not find Ω to strongly depend on canopy height in both observation and simulation. Although the highest canopy tends to have positive SHAP values, the range of SHAP values for smaller height levels is very large with both positive and negative.

A comparison of all controlling factors individually between the observations and the ORCHIDEE simulations is shown in
Fig. 6. The dependence of Ω on wind speed generally has similar patterns in observation and in ORCHIDEE. However, when
Ω is decomposed to Ga and Gs, differences between observation and ORCHIDEE appear. According to the observation, when
wind speed is smaller than 5 m s⁻¹, an increase of wind speed will contribute to larger Ω, while when wind speed is larger than
5 m s⁻¹, increase of wind speed will not further affect Ga significantly. In contrast, ORCHIDEE simulations show an increase
of Ga continuously with wind speed at large wind speeds. In observation, we also found positive SHAP values of wind speed
at wind speed smaller than 1 m s⁻¹, this might be due to coincidence because low wind speed will cause large uncertainty in
the eddy covariance measurements and there are very few valid observation-based Ω available at low wind speed.

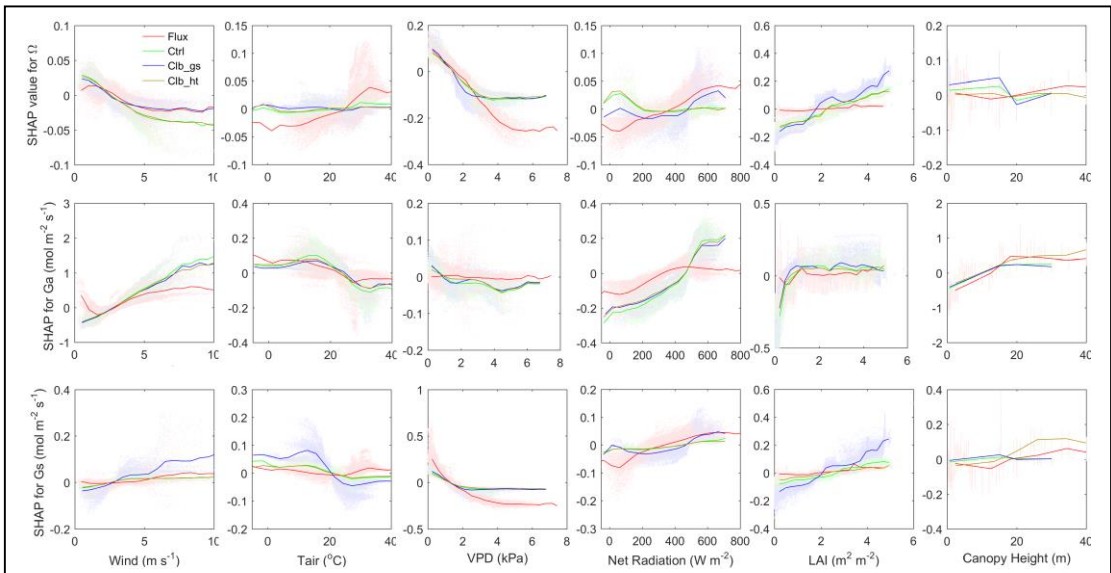

Figure 6. Dependence of Ω (top), Ga (middle) and Gs (bottom) SHAP values on different factors (in
order from left to right: wind speed, air temperature, VPD, net radiation, LAI and canopy height).
The colors indicate observation or simulations. red: observation-based dataset, green: Ctrl, blue:
Clb_gs, brown: Clb_ht. The shaded dots show the distribution of SHAP values in sample.

The observed dependence of Ω on Tair is not captured by ORCHIDEE. Observations indicate an increase of Ω when Tair is
lower than 30°C, and a slight decrease at higher temperature. While ORCHIDEE simulations show a much smaller impact
from Tair. The model bias is caused by differences in the relationships of Gs on Tair at high temperature. A strong decline of
the Gs SHAP values is found when the Tair is over 20°C in ORCHIDEE, while the observations show a slight increase of Gs
SHAP values at the same temperature. This difference probably indicates an underestimation of optimal temperature for
photosynthesis in ORCHIDEE in PFTs that have been acclimated to hot weather.

In terms of the VPD, ORCHIDEE generally captures the negative dependence of Ω to VPD at VPD smaller than 2 kPa.
However, when the VPD is larger, observations show continuous negative dependence of Ω, while ORCHIDEE simulations





show no significant changes in Ω with VPD. The decomposition into components of Ω shows that this difference is mainly contributed by different dependence of Gs on VPD (Fig 6).

Compared with the observations, ORCHIDEE simulations show a different dependence of Ω to Rnet when the net radiation is <100 W m⁻². This difference is also mainly contributed by differences in Gs. In observation, the Gs SHAP values start to

decrease rapidly when Rnet is lower than 200 W m⁻², while in ORCHIDEE simulations, the decrease of SHAP values is smaller and happens when Rnet is below 50 W m⁻².

Regarding the dependence of Ω to LAI, ORCHIDEE simulations show a significant increase of Ω with LAI across the entire range of LAI, due to a strong increase of Gs along with LAI, with the Gs SHAP values increasing by 0.2-0.4 mol m⁻² s⁻¹ from LAI=0 to LAI=5. However, the observations show that SHAP values increase only by less than 0.05 mol m⁻² s⁻¹ for the same

change in LAI, resulting in a weak dependence of Ω on LAI.

Both observation and ORCHIDEE show weak dependence of Ω on canopy height. However, all of the data agree with a positive impact of canopy height on Ga. A strong increase of Ga is found when the height is below 15 m.

### 3.3 Interactions among factors

To further understand how the model biases in the controls of Ω, we explored the interactions between factors that have

significant different impacts between ORCHIDEE and observations (Fig 7, 8).

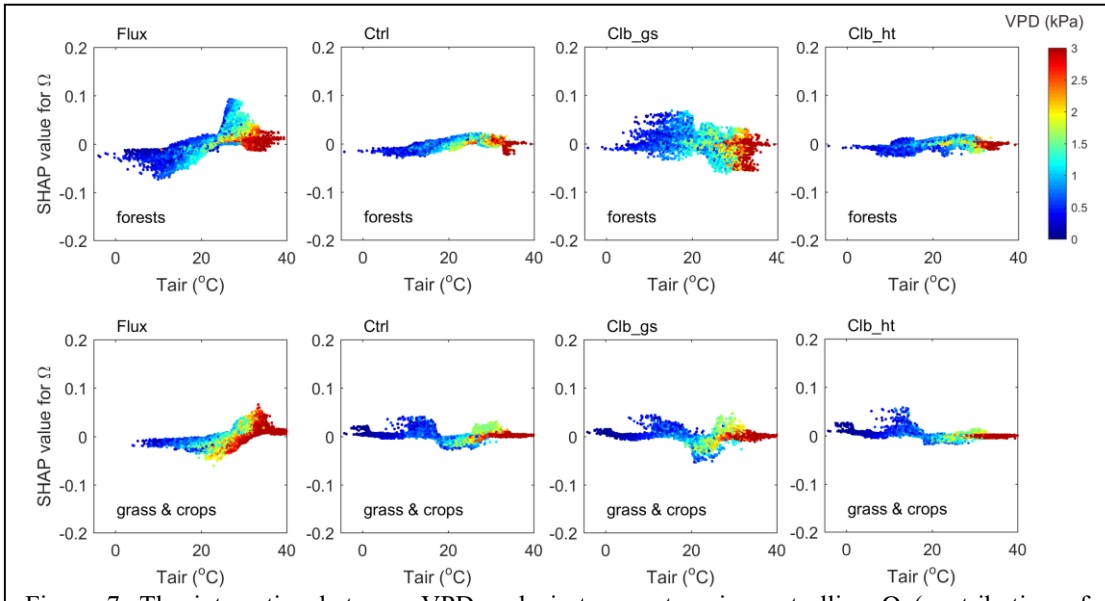

Figure 7. The interaction between VPD and air temperature in controlling Ω (contribution of temperature) in forests (top) and in grasslands and croplands (bottom). The y axis is the SHAP value of Tair for Ω, colors indicate the VPD of each data point.

The interactions between VPD and Tair are shown in Fig. 7. The observation data show that when Ω SHAP value is positive (Tair >25°C), data with larger VPD have smaller Ω values than those with smaller VPD.



In ORCHIDEE simulations, although Ω SHAP values varies differently along the temperature gradient compared with observations, similar interactions between VPD and Tair are also found, i.e., for a given temperature when Ω SHAP value is
positive, large VPD values tend to result in smaller Ω. In another words, the dependence of Ω to Tair in hot weather is weakened by high VPD level. This weakening of Ω dependence on Tair is due to weakened dependence of Gs on Tair under high VPD conditions (Fig S3).

A similar interaction between VPD and LAI is also found in both the observations and ORCHIDEE simulations (Fig 8). The data points with VPD>3kPa show SHAP values close to zero, indicating that higher VPD tends to also weaken the dependence
of Ω on LAI. ORCHIDEE underestimated the weakening effect of high VPD to the Ω to LAI dependence as the SHAP values under high VPD conditions remain very positive/negative compared with the observation.

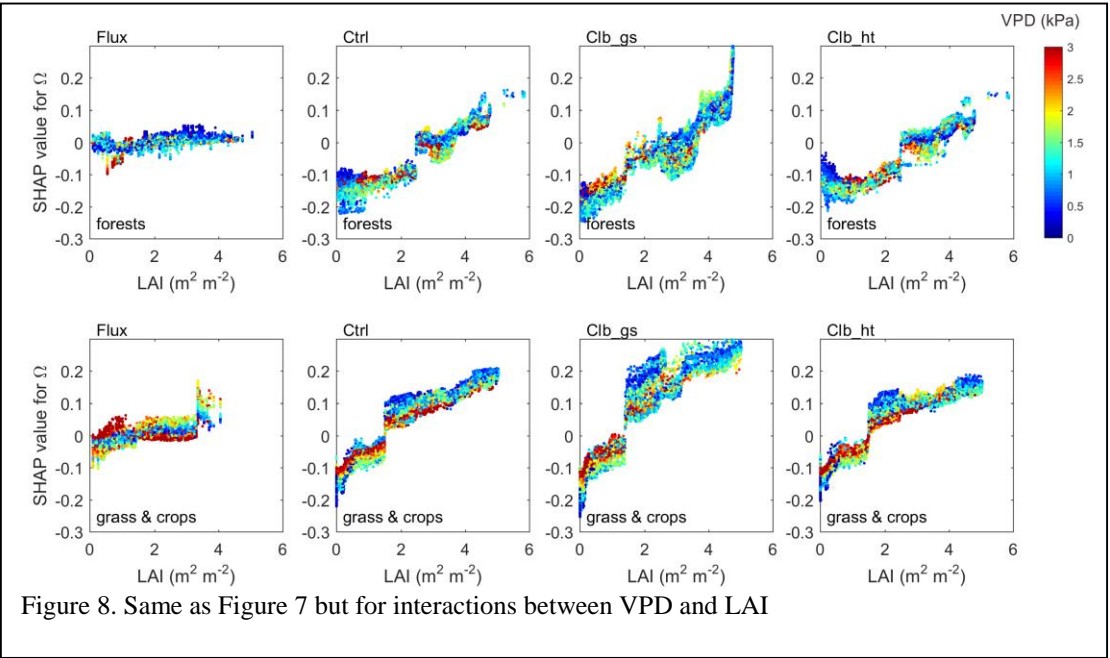

Figure 8. Same as Figure 7 but for interactions between VPD and LAI

## 4 Discussion

### 4.1 How can models correctly simulate the decoupling strength

Accurately resolving the land-atmospheric water/energy exchanges is critical in simulating the climate system. To ensure this,
LSMs must be carefully calibrated and validated with observations before use. The ORCHIDEE model has been calibrated several times for carbon and water fluxes against flux observations including the use of dedicated data assimilation systems (e.g., Bastrikov et al., 2018). As a result, the ORCHIDEE model with the most recent set of parameters does not show large biases in LE (Fig 3c).





Nevertheless, there remains no evaluation on the components and processes of LE, as well as their biotic and abiotic controls,
leading to potential biases in LE simulation if climate changes. Disentangling and assessing processes and components of LE
are difficult due to the lack of direct observation (Nelson et al., 2020). Although not perfect, evaluating the coupling strength
and its components gives a possible way to further constrain the models.

In this study, we showed that current ORCHIDEE model captures the coupling strength at most of the sites, however fails to
correctly represent the processes. The tuning of current LSM models often adjusts a few uncertain parameters to produce a
small number of target variables (C fluxes, LE, sensible heat flux) close to the observation. In a complex model, this kind of
calibration may result in overfitting, and errors compensating for each processes. In the end, the model may get the correct
result for the wrong reasons. Therefore, calibration the model at the process level is helpful. For instance, the calibration of
the $a_1$ and $b_1$ parameters in stomatal conductance calculation using independent observation-constrained values from Lin et al.
(2015) leaf scale data synthesis has significantly improved our estimation of fvpd (Fig. S1), and consequently corrected some
biases in Gs and resulted in better $\Omega$ in short vegetation. In Forest sites, $\Omega$ seems worse after this calibration, but this is because
of the biases in modeled Ga, probably due to bad assumption in calculating the displacement height.

In spite of the improvement from gs calibration, there remain large biases in Gs in short vegetation (grasslands and croplands).
Our analyses on the controlling factors sheds light on where the problems are and give a direction to improve: we expect the
model performance to improve if the dependence of Gs on temperature is corrected and the impact of VPD on stomatal
conductance is further constrained. We did not do further calibration here because the responses of gs to VPD are an emergent
area of concern for LSMs and more process-level modeling and calibration efforts remain needed (Yang et al., 2019). Also, it
is out of the scope of this evaluation study. Nevertheless, the framework we used here would be helpful for models to identify
their problematic processes and potentially fix their biases.

## 4.2 Factors controlling vegetation coupling strength

Due to the complexity of processes, as well as the lack of data, it is difficult to attribute the variation of coupling strength to
different factors. Previous studies either focus on one or a few meteorological factors such as VPD, radiation or wind speed
(Kumagai et al., 2004; Nicolás et al., 2008; Zhang et al. 2018) or biotic factors like LAI or PFT (Tateishi et al., 2010; Zhang
et al., 2016). Our new framework to disentangling the impacts of different factors provides a systematic view to understand
the impact of these factors.

Among all the factors, VPD was the most intensively investigated factor due to its strongest impact on stomatal conductance.
Previous study showed that vegetation tends to be more decoupled in wet season with low VPD compared with dry season
with high VPD (Kumagai et al. 2004). In this study, we found VPD the most important factor affecting $\Omega$ and to affect $\Omega$
similarly as the previous study (Fig 6). This effect is mainly due to the reduction of Gs under dry conditions as plants tend to
close the stomata under high VPD conditions to reduce water loss. In addition, high VPD conditions often coincide with low
soil moisture, which hampers soil water uptake by plants, also leading to low Gs. It should be noted that this VPD-$\Omega$





relationship is obtained using daily data. At sub-daily time scale, this VPD-$\Omega$ relationship is not easily observed due to the strong impacts of other factors (Wullschleger et al., 2000; Zhang et al., 2018).

The impact of Tair on $\Omega$ is through two possible pathways. First, Tair can directly affect VPD by changing saturate water vapor pressure, leading to changes in $\Omega$. Second, Tair can affect the photosynthesis rate by changing enzyme activities. Because
stomatal conductance is strongly coupled with carbon assimilation rate (Cowan and Farquhar, 1977), the changes in photosynthesis rate can thus affect gs, and consequently $\Omega$. In this study, we found that the responses of $\Omega$ and Gs to Tair different from those to VPD, implying that the impacts of Tair through the second pathway is not negligible. The differential Tair impacts on Gs and $\Omega$ between observation and model simulations are probably due to wrong Tair adaptation of vegetation in ORCHIDEE model.

Besides VPD and Tair, some studies found significant impacts from net radiation (Nicolás et al., 2008) or photosynthetically active radiation on $\Omega$ (which is strongly correlated to net radiation used in our analyses) (Zhang et al., 2018). Similar to Tair, changing radiation can also alter leaf photosynthesis rate. Due to the coupling between stomatal conductance and carbon assimilation, the changes in radiation thus result in $\Omega$ changes. Nevertheless, the impact of radiation should be considered with caution because radiation is strongly correlated with other environmental or biotic factors that have diurnal and seasonal cycles
(e.g. temperature, LAI). Besides the short-term effect, long term changes of radiation can affect soil moisture by altering LE, which may potentially change the coupling strength of the vegetation.

In terms of wind speed, we detected a negative dependence of $\Omega$ on wind as expected. This is because wind can accelerate the mixing of the boundary layer, increasing Ga. In this study, we did not find wind speed to be as important as VPD or vegetation types in explaining the variation of $\Omega$. However, it needs to be kept in mind that the importance of factors depends on vegetation
type. In ecosystems with a small vegetation cover (meaning small Gs), or in ecosystems where Gs has small variability, the importance of wind speed will increase.

Apart from the abiotic factors, the biotic factors, or vegetation properties also play important roles in controlling $\Omega$. The PFT is found the second important factor affecting $\Omega$ after VPD in observation data (Fig 4). In ORCHIDEE simulations, the PFT impact on $\Omega$ is weaker but still important, especially for different forest types. The pattern of $\Omega$ among PFTs found in this
study agree well with De Kauwe et al. (2017). The influences from PFT types on $\Omega$ may be due to various reasons. Besides leaf area and canopy height (investigated in this study), different PFTs often have different canopy structure and leaf traits, leading to differences in Ga and Gs. Meanwhile, the climate and environmental conditions (e.g. soil types) which different PFTs adapted to are also different. More detailed data are needed to further explain the PFT impacts.

In the two biotic factors, canopy height is thought to be important factor in affecting $\Omega$ because it directly affects the roughness
length and the aerodynamic resistance (Ershadi et al., 2015). Higher canopies with larger roughness tend to enhance the turbulence for a given wind speed above the canopy. In this study, we found a positive but weak dependence of Ga on canopy height when the height is under 15m. This result is consistent with Peng et al. (2019), who found that when controlling leaf area, $\Omega$ decreases (corresponding to Ga increase) with canopy height in vegetation with height<20m. In higher canopies, Ga and $\Omega$ becomes less sensitive to canopy height.





Besides canopy height, LAI is also an important control. On the one hand, observations have shown that large LAI can increase the roughness (Alekseychik et al., 2017), which can lead to an increase of Ga. Along with LAI, leaf size might be also important in affecting the roughness and Ga, but is not available at most sites, neither simulated by ORCHIDEE model. On the other hand, LAI affects Gs as a larger LAI means a larger area for transpiration. This effect might be further regulated by environmental factors such as VPD (Fig 8). Besides the influence from environmental factors, we also expect the impact of

LAI on Gs to saturate for high LAI, because of increasing self-shading. The shaded leaves in lower canopy tend to have smaller transpiration due to the low interception of radiation (Roberts et al., 1993), resulting in a decrease of average transpiration per leaf area. Also, the Gs at the ecosystem level is a synthesis of different processes including the vapor diffusion within the canopy. Large LAI may slow down the diffusion of water vapor within the canopy, potentially resulting in smaller Gs, and smaller $\Omega$.

## 4.3 Limitation

Although the simulations and analyses we performed in this study clearly showed how and why ORCHIDEE LSM has biases in its estimation of the decoupling strength, there remain some questions which need to be answered before we can calibrate the processes underlying these biases.

First, the decoupling strength is the consequence of multiple processes. In this evaluation of $\Omega$, strict criteria have been used

to screen the data to have only time steps with LE mainly contributed by transpiration. The effect of other processes (e.g. soil evaporation) can potentially affect the decoupling strength under some circumstances. For instance, the wetland $\Omega$ is also strongly affected by evaporation from open water. An understanding of these processes is also important, and our evaluation cannot draw conclusions on how well ORCHIDEE simulates these processes.

Second, due to the meteorological requirements of eddy covariance methods, the current selected observations have an

incomplete coverage of the real meteorological conditions. We could not obtain valid observations under conditions with no wind. However, plants still transpire water to the atmosphere under such conditions. New observation methods are needed to fill this gap.

The data used in this study are all day-time values. But for some vegetation types, transpiration also happens at nighttime (Dawson et al., 2007). Although the nighttime transpiration is smaller than the day-time transpiration, it can still affect the

water and energy balance at longer time scale. These changes can potentially affect vegetation. However, the processes controlling the nighttime transpiration, as well as how coupled the ecosystems are at night remains poorly understood. Current LSMs also lack representations of such processes. We are not able to consider these processes in our evaluation/simulation.

Besides the missing processes, uncertainty may also come from the method to estimate $\Omega$. In the observation-based estimates, Ga was estimated using an empirical method from Thom et al. (1975), which has inevitable uncertainties. Nevertheless,

estimates from this method are found to be consistent with other more physically based methods (Knauer et sl., 2017). For Gs, the inverted Penman-Monteith equation may also result in some uncertainties. On the one hand, the energy budget is not always closed in flux observations. De Kauwe et al. (2017) used the value zero when soil heat flux observation is absent in estimating





Gs, which could lead to biases in Gs and consequently Ω if the actual soil heat flux is not negligible. On the other hand, Penman-Monteith equation remains not perfect in estimating LE. A recent study (McColl, 2020) showed that the linear approximation of Clausius-Clapeyron relation in the Penman-Monteith equation may cause significant biases when there is large difference between ambient air temperature and surface temperature (often with small Ga). Using inverted Penman-Monteith equation with observed LE may thus bias the Ω estimates. However, since ORCHIDEE used the same method to estimate Gs as the observation, the uncertainties from the Penman-Monteith equation should not significantly affect our findings and conclusion.

## 5 Conclusion

In summary, in this study we evaluated the vegetation-atmosphere coupling strength, Ω, in ORCHIDEE LSM using an observation-based dataset at 106 flux sites. We found that short vegetation (grassland and cropland) in ORCHIDEE is too tightly coupled to the atmosphere compared to the observation-based estimates, while the coupling strength of forests is generally well estimated by ORCHIDEE. Nevertheless, there remains biases in both modeled Ga and Gs. Calibration of parameters controlling the dependence of the stomatal conductance to VPD reduces the biases of Gs in ORCHIDEE model to a small extent and improves the Ω estimates in short vegetation. Using a set of random forest models and analyses on SHAP values, we found that vegetation tends to be more decoupled to atmosphere at low wind speed, high temperature, low VPD and large LAI conditions and in short vegetation. ORCHIDEE generally agrees with this pattern but underestimated the VPD impacts when VPD is high, overestimated the contribution of LAI and did not correctly simulate the temperature dependence when temperature is high. Canopy height affects Ga but does not show a strong direct impact on Ω. Our results highlight the importance of observational constraints on simulating the vegetation-atmosphere coupling strength, which can help improve the predictive accuracy of water fluxes in Earth system models.

### Code availability

The ORCHIDEE model code is available at https://forge.ipsl.jussieu.fr/orchidee/wiki/GroupActivities/CodeAvalaibilityPublication/ORCHIDEE_2.2_gmd_2022.

### Data availability

The Ω data used in this study is from De Kauwe et al. (2017). And the FLUXNET data is obtained at https://fluxnet.org (Pastorello et al., 2020)



**Author contribution**

YZ and DN performed the simulations and analyses, MDK estimated $\Omega$ at fluxnet sites. YZ prepared the manuscript with the contributions from all the co-authors.

**Acknowledgements**

The authors are grateful to the ORCHIDEE group for their kind help with the model. The authors are very grateful to the FLUXNET communities for their efforts with respect to making the sites and collecting data. The authors also thank Dr Xiaoni
Wang for providing the ORCHIDEE PFT fraction at each flux site.

**Financial support**

This study is supported by the H2020-EU.3.5.1. 4C project (grant no. 821003). DG benefited from support from the Agence Nationale de la recherche (ANR) grant ANR- 16- CONV- 0003 (CLAND).

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
