# Peer review of "Evaluating the vegetation-atmosphere coupling strength of ORCHIDEE land surface model (v7266)"

_Geoscientific Model Development, 2022_

## Referee Comment (RC1)

This paper evaluates the representation of the decoupling coefficient in an ESM model. This is an important observational based study with a lot of discussion on the model processes and it will contribute to the model development of evapotranspiration process. The study also uses a machine learning approach to rank the controlling factors, which is a new alternative of regression coefficients. The article structure is clear but there are quite a few grammar errors in the writing, please fix them for readability.

Title: coupling strength is used in the title. However, coupling strength is the reverse of Omega, the main concept of the paper. It could be confusing to readers when both coupling strength and decoupling strength are used. I suggest use "decoupling strength" consistently. You can still say "coupling".

Paragraph 1-2: Lack of literature support. Please add more references for each statement.

L44: improve the simulation **of** transpiration

L59: total evapotranspiration (e.g., Peng et al. 2019)

Equation 6: how is the Ga estimated when calculating the empirical Omega? Not clear whether this study is following Thom et al. 1975, equation 3, or taking from the De Kauwe dataset? I am afraid the selection of this formula will greatly affect your results of Gs, Omega, and the following analysis.

Figure 3a shows a systematic large gap of Ga between Flux and Ctrl. There are uncertainties in both data and the parameterization. When you compare it with ORCHIDEE, have you validated if the modelled Ga is consistent the empirical Ga when you use the same ORCHIDEE formula?

Also the equations (4)-(5) are highly dependent on vegetation structure. I am afraid this is not going to work very well across biomes. Can you do a sensitivity test of the formula across biomes?

L140: The Gs in the equation has the "s" as subscript, but the text is different. Similar in other places.

Fig 6 – the scales of yaxis are different and difficult to compare.

L255: the response of Omega to temperature is nonlinear because VPD also depends on temperature. Does this RF reflect such nonlinear relationship?

---

## Author Comment (AC1)

**Response to Reviewer #1**

**Comments:**

*This paper evaluates the representation of the decoupling coefficient in an ESM model. This is an important observational based study with a lot of discussion on the model processes and it will contribute to the model development of evapotranspiration process. The study also uses a machine learning approach to rank the controlling factors, which is a new alternative of regression coefficients. The article structure is clear but there are quite a few grammar errors in the writing, please fix them for readability.*

**[Response]** We thank the Reviewer for the positive comments and helpful suggestions. We have addressed all the suggestions and comments in our revision. Please find below the Reviewer's comments (italics), followed by our responses (roman font), with red color indicating relevant changes in the manuscript. We hope that the revised manuscript addresses all the issues raised by the Reviewer.

*Title: coupling strength is used in the title. However, coupling strength is the reverse of Omega, the main concept of the paper. It could be confusing to readers when both coupling strength and decoupling strength are used. I suggest use "decoupling strength" consistently. You can still say "coupling".*

**[Response]** Thanks for this suggestion, we had a discussion on the title and we thought that decoupling coefficient is a proxy to measure the coupling between vegetation and the atmosphere. It is named "decoupling" because of its large values corresponding to smaller coupling strength. We used "coupling strength" rather than "decoupling strength" in the title so that readers who are not familiar with this coefficient can better understand. To avoid confusion, we have revised the manuscript to use only "coupling strength" and "decoupling coefficient" in the manuscript.

*Paragraph 1-2: Lack of literature support. Please add more references for each statement.*

**[Response]** Thanks for this suggestion. We have added more references to support our statements.

*L44: improve the simulation **of** transpiration*

*L59: total evapotranspiration (e.g., Peng et al. 2019)*

**[Response]** They have been corrected accordingly

*Equation 6: how is the Ga estimated when calculating the empirical Omega? Not clear whether this study is following Thom et al. 1975, equation 3, or taking from the De Kauwe dataset? I am afraid the selection of this formula will greatly affect your results of Gs, Omega, and the following analysis.*

**[Response]** Sorry for the confusion. In this study, the reference data (flux) is from the De Kauwe dataset, in which Ga was estimated following Thom et al. (1975). In the model, Ga is calculated using Eq 3. We agree that the different formulation for Ga can result in biases, but we don't have enough information to estimate Flux Ga using the same equations as in ORCHIDEE. To investigate the impact of the Ga estimation, we performed a sensitivity test by perturbing Ga by 30% (see the response to the 2nd comment of Reviewer 2). We found that decreasing Ga by 30% may result in smaller model bias in Ga, but the change of Ga does not alter the overall pattern of omega and Gs across PFTs and the dependence of omega to different factors, thus not affecting our conclusions. We added a discussion on the impact of empirical omega uncertainties. Lines 625-634: "In the observation-based estimates, Ga was estimated using an empirical method from Thom et al. (1975), which was derived from a bean crop. Ga estimates from this method are found to be 81%-116% of the estimates of a more physically based method (Knauer et sl., 2017) in 6 forest sites. To test how biased Ga affects our evaluation, we increased/decreased Ga by 30% and re-estimated Gs and $\Omega$ (Fig S6). We found that perturbing Ga does not result in large changes in Gs. However, when Ga is 30% smaller than current observation-based estimates, we obtained smaller biases in Ga and $\Omega$ in ORCHIDEE Ctrl simulation in forest PFTs. Whereas in short PFTs, decreasing the reference Ga results in even larger biases in $\Omega$, indicating that the large biases in model vegetation coupling strength in short vegetation is not due to uncertainties in the observation-based estimates."

*Figure 3a shows a systematic large gap of Ga between Flux and Ctrl. There are uncertainties in both data and the parameterization. When you compare it with*

*ORCHIDEE, have you validated if the modelled Ga is consistent the empirical Ga when you use the same ORCHIDEE formula?*

**[Response]** We also noticed the big difference between modeled and observed Ga. The comparison proposed by the reviewer would be helpful to understand the bias, however, some variables used in ORCHIDEE formula are not available at flux sites (e.g. $z_{0h}$, $z_{0m}$). To further investigate the reason of this large gap, reliable estimates of these variables are needed to evaluate the parameterization of $z_{0h}$ and $z_{0m}$ in the model.

*Also the equations (4)-(5) are highly dependent on vegetation structure. I am afraid this is not going to work very well across biomes. Can you do a sensitivity test of the formula across biomes?*

**[Response]** The calculation of $z_{0h}$ and $z_{0m}$ in all PFTs follow the same equation in ORCHIDEE model. The calculation depends on LAI and canopy height. Following the reviewer's suggestion, we performed a sensitivity test of z0h and z0m across different LAI and canopy height (Fig R1) and have put the result in the supplementary. We also added a more detailed description of these formula to the manuscript: Lines 162-177: "

$z_{0m}$ and $z_{0h}$ are respectively the roughness heights (m) for momentum and heat transfer estimated following Su et al. (2001) and Ershadi et al. (2015) using canopy height ($z$) and LAI:

$$z_{0m} = (z - d)e^{-\frac{\kappa}{\eta}} \qquad (4)$$

Where

$$\eta = 0.32 - 0.264e^{-3.02LAI} \qquad (5)$$

$z_{0h}$ is estimated using $z_{0m}$ :

$$z_{0h} = \frac{z_{0m}}{e^{\kappa B^{-1}}} \qquad (6)$$

B is the Stanton number. $\kappa B^{-1}$ is estimated following Su et al. (2001; 2002):

$$\kappa B^{-1} = \frac{\kappa Cd}{4Ct\eta(1-e^{-\frac{nec}{2}})}fc^2 + 2fcfs\frac{\kappa\eta\frac{z_{0m}}{z}}{C_t^*} + \kappa B_s^{-1}fs^2 \qquad (7)$$

Where Cd, Ct are drag and heat transfer coefficient of leaves, nec is within canopy wind profile extinction coefficient, calculated as nec = CdLAI/($2\eta^2$). fc, fs are the fraction of canopy and bare soil, $C_t^*$ is the heat transfer coefficient of soil. Bs is the Stanton number for bare soil, with $\kappa B_s^{-1}$ estimated following Brutsaert (1999):

$$\kappa B_s^{-1} = 2.46Re_*^{\frac{1}{4}} - \ln(7.4) \qquad (8)$$

Where Re* is the Reynolds number."

[Figure]

Figure R1. The dependence of Ga on LAI and different canopy heights in ORCHIDEE parameterization under 3 m s$^{-1}$ wind speed, sea level pressure and 15 ℃ condition

*L140: The Gs in the equation has the "s" as subscript, but the text is different. Similar in other places.*

**[Response]** Thanks for this remark, we have unified the symbols in the equations and the text.

*Fig 6 – the scales of yaxis are different and difficult to compare.*

**[Response]** The y axes are now unified.

*L255: the response of Omega to temperature is nonlinear because VPD also depends on temperature. Does this RF reflect such nonlinear relationship?*

**[Response]** Yes, the RF model is able to deal with nonlinear relationships, as it regresses data by grouping but not by fitting lines. However, RF is still a statistical method, it cannot fully decompose the impact of factors that have strong dependence relationships. If there are factors with very strong dependence, the RF algorithm may randomly select one of these factors to split the trees, resulting in non-robust results. In this study, we found robust contributions of VPD and temperature in the RF models built with different data (Fig 5), indicating that our result is not an artifact due to the dependence between factors.

---

## Author Comment (AC2)

**Response to Reviewer #2**

[Response] We thank the Reviewer for the helpful comments and suggestions, which improved our manuscript significantly. We have addressed all the suggestions and comments in our revision. Please find below the Reviewer's comments (italics), followed by our responses (roman font), with red color indicating relevant changes in the manuscript. We hope that the revised version addresses all the issues raised by the Reviewer.

*In this manuscript, Zhang et al. compare the atmosphere-vegetation decoupling factor omega simulated by the land surface model ORCHIDEE with observation-based estimates derived from 106 eddy covariance sites across PFTs. The motivation of this study is valuable as it attempts to evaluate the model performance of simulating an ecosystem property rather than just the simulation of fluxes themselves. This is useful as it allows deeper insights into the functioning of the model compared to more common evaluation approaches. However, I found one critical point that needs to be corrected before the article can be published. In Eq. 3, z should be the reference height (or sensor/measurement height for the eddy covariance towers), not the vegetation height. See e.g. the Monteith and Unsworth 2013 textbook (4th edition, Eq. 17.6) or Liu et al. 2007. To be consistent with the tower observations, z must be set to the height of the sensor at the flux sites in the model. If this is not done a series of biases are introduced which affect the rest of the results. For example, the interpretation of a measured wind speed at a flux site depends on the height where it's measured (logarithmic increase with height), but in the model the assumption is made that the wind speed was always measured at vegetation height, which is incorrect and which causes biases in Ga that affect different PFTs to a different extent.*

[Response] Thanks for pointing out this problem in our method. To address this problem, we checked out the FLUXNET dataset and re-run all our simulations at site where the measurement height is available (Table S1). In the new simulations, we kept the distance between canopy top and the measurement height consistent with the observations (Eq. 3). All the results in the manuscript are updated accordingly, using the new simulation outputs. The description of the new simulations has been added to

the revised manuscript, Lines 266-269: "In all the simulations, we kept the distance between measurement height and canopy height consistent with the observations, to ensure unbiased estimates of aerodynamic conductance in the model."

Our updated results show that the biases in current ORCHIDEE model are not from the biases in measurement height, but from model processes.

*A second major point is that more care must be taken in how 'observations' of omega are used in such an evaluation exercise. The omega values here are modelled products as well (observation-based at best) which come with a range of assumptions that will affect its interpretation. For example, the canopy boundary layer conductance in de Kauwe et al. 2017 from Thom et al. 1972 was derived for a bean crop but applied to all PFTs in that study. That will inevitably lead to biases in the flux-derived omega values affecting different PFTs to a different extent. Similar issues arise from a non-closure of the energy balance that lead to negative biases in Gs (probably also different in different biomes/PFTs). The authors point to these issues in the discussion, but it would be most useful to the reader if the consequences of these potential biases on the results were elaborated. Just describing that an issue causes a bias is not very useful. It would be much better to show what factors are likely to have major impacts on the comparison by conducting e.g. sensitivity analyses. Knowing that a model-data mismatch could be caused by either the observations or the model is absolutely crucial for such an analysis.*

**[Response]** Thanks for this suggestion. Following the reviewer's suggestion, we tested how biases in Ga and Gs affect our results (Fig R2).

First, we tested how increasing and decreasing observation-based Ga by 30% affects Gs and omega. We found that perturbing Ga does not result in large changes in Gs. However, using a reference Ga that is 30% smaller than current observation-based estimates, we obtained a better consistency between Ga and omega from ORCHIDEE Ctrl simulation and the reference in forest PFTs. Whereas in short PFTs, decreasing reference Ga to the level of model output (-30%) results in even larger biases in omega, indicating that the large biases in model vegetation coupling strength in short vegetation is not due to uncertainties in the observation-based estimates.

Then we tested how large the energy imbalance affects our evaluation. Using the same method as in De Kauwe et al. (2017), we corrected the energy imbalance and recalculated Gs and omega. The model shows larger Gs biases across all PFTs when

[Figure]

Figure R2. Impacts of uncertainties in the empirical calculation of Ω on the comparison. The boxes from the left to right: Ctrl simulation, De Kauwe et al. (2017) dataset, increasing Ga by 30%, decreasing Ga by 30%, correction of the energy imbalance.

compared with the estimates after correction. Although, the biases in Gs compensate to the existing biases in Ga and result in good performance of omega simulation in forest PFTs.

We have added the new results and discussion to the revised manuscript. Lines 625-634: "In the observation-based estimates, Ga was estimated using an empirical method from Thom et al. (1975), which was derived from a bean crop. Ga estimates from this method are found to be 81%-116% of the estimates of a more physically based method (Knauer et sl., 2017) in 6 forest sites. To test how biased Ga affects our evaluation, we increased/decreased Ga by 30% and re-estimated Gs and Ω (Fig S6). We found that perturbing Ga does not result in large changes in Gs. However, when Ga is 30% smaller than current observation-based estimates, we obtained smaller biases in Ga and Ω in ORCHIDEE Ctrl simulation in forest PFTs. Whereas in short PFTs, decreasing the reference Ga results in even larger biases in Ω, indicating that the large biases in model vegetation coupling strength in short vegetation is not due to uncertainties in the observation-based estimates."

Lines 641-645: "When the energy imbalance is corrected by adjusting the Bowen-ratio following De Kauwe et al. (2017), we obtained larger Gs estimates (Fig S6), resulting in even larger modeled Gs bias than in this study. The increased biases in the corrected Gs compensate for the existing biases in Ga, leading to a "good" performance of omega simulation in forest PFTs."

**Minor comments:**

**Introduction:**

*l. 43: add 'of' after simulation*

*l. 45: one cannot add conductances if they are in series, only resistances. Please rephrase.*

**[Response]** Corrected

*L. 52: surely there must be more references supporting this statement.*

**[Response]** Thanks for this suggestion, we have added more references to the manuscript.

**Methods:**

*L100: please describe how stomatal conductance (gs) is scaled to the canopy level.*

**[Response]** The description of the stomatal conductance is added. Line 153-154: "The canopy level stomatal conductance is calculated by integrating gs across all leaves in the canopy."

*L 115: please give the equation for z0h here, not only the reference. The z0h to z0m ratio is relevant for the canopy boundary layer conductance and thus Ga. Does the equation imply differences in z0h/z0m across PFTs?*

**[Response]** We have added the equations to the manuscript.

Lines 168-177: "$z_{0h}$ is estimated using $z_{0m}$ :

$$z_{0h} = \frac{z_{0m}}{e^{\kappa B^{-1}}} \qquad\qquad (6)$$

B is the Stanton number. $\kappa B^{-1}$ is estimated following Su et al. (2001; 2002):

$$\kappa B^{-1} = \frac{\kappa C d}{4 C t \eta (1-e^{-\frac{nec}{2}})} fc^2 + 2fcfs \frac{\kappa \eta \frac{z_{0m}}{z}}{C_t^*} + \kappa B_s^{-1} fs^2 \qquad (7)$$

Where Cd, Ct are drag and heat transfer coefficient of leaves, nec is within canopy wind profile extinction coefficient, calculated as nec $= CdLAI/(2\eta^2)$. fc, fs are the fraction of canopy and bare soil, $C_t^*$ is the heat transfer coefficient of soil. Bs is the Stanton number for bare soil, with $\kappa B_s^{-1}$ estimated following Brutsaert (1999):

$$\kappa B_s^{-1} = 2.46 Re_*^{\frac{1}{4}} - ln(7.4) \qquad\qquad (8)$$

Where Re* is the Reynolds number."

The *z0h/z0m* is affected by LAI and canopy height, thus implicitly different across PFTs. In current ORCHIDEE version, the parameters to calculate *z0h/z0m* is the same for different PFTs.

*L 120: add 'radiation' after short-wave down and long-wave down*

**[Response]** Added accordingly.

*Where in Su et al. 2001 can Eq. 4 be found?*

**[Response]** Eq. 4 in this manuscript is derived from Eq. 10 in Su et al. (2001)

*Section 2.2 is inaccurately named as there is basically no information on the FLUXNET data used. I argue that this section needs to be split up: one part giving more*

*information on the data used (e.g. what sites from what biomes), which could be combined with section 2.3, and then a separate section for the simulation setup.*

**[Response]** Thanks for this suggestion, we have split this section as the reviewer suggested. Now Section 2.2 is "FLUXNET data and empirical calculation of Ω", and Section 2.3 is "Simulation setup and modeled Ω calculation".

*L 152ff: I do not understand why the integrated canopy level stomatal conductance was not used? Surely the canopy-integrated value needs to be used in order to be comparable with the flux-derived omega.*

**[Response]** Here we used the Gs calculated from inverted Penman–Monteith equation because the latent heat flux simulated by ORCHIDEE includes bare soil evaporation and evaporation from the rainfall intercepted by the canopy. Also the observation-based estimates of Gs implicitly include these processes. To be consistent with the reference, we used the same way to calculate modeled Gs.

*L. 158: sites*

*L. 209: observations*

*L. 214: compensation*

**[Response]** Revised accordingly.

***Discussion***

*L 327: what other factors? Please explain.*

**[Response]** The diurnal change of radiation may strongly affect the coupling strength. We added this into the manuscript. Line 532-534: "At a sub-daily time scale, this VPD-Ω relationship is not easily observed due to the strong impacts of other factors, such as radiation (Wullschleger et al., 2000; Zhang et al., 2018)."

*L. 370: Limitations*

**[Response]** Corrected.

*L. 380: I find this point a bit senseless. If conditions with wind speed don't occur across the data set, why should these conditions matter? They may be of interest from a theoretical point of view but not for LSM evaluation.*

**[Response]** Thanks for this comment. Here we discuss this point because we think that our framework is not only helpful to understand the bias but also has the potential to

calibrate the model. However, there remains observation gaps. In the manuscript, we emphasized this point. Line 612-614: "New observation methods are needed to fill this gap so that future calibrations can ensure the models to correctly simulate vegetation under all different conditions."

*L. 389: what are the uncertainties and how would this affect observation-derived omega values?*

**[Response]** Please see the response to the 2nd major point

*L. 397: how and under which conditions would this bias the results? This needs to be explained better.*

**[Response]** The bias is explained in the revised manuscript. Lines 649-652: "A higher surface than ambient air temperature (daytime) tends to overestimate Gs in the inverted Penman-Monteith equation with observed LE, which can further overestimate omega."

---

## Author Response (AR2)

**Response to Reviewer #2**

*In their revision of the paper entitled "Evaluating the vegetation-atmosphere coupling strength of ORCHIDEE land surface model (v7266)", the authors have addressed all of the major comments I have raised in the first review. Major updates are 1) a revision of the Ga calculation in ORCHIDEE, in particular replacing vegetation height with measurement height, and 2) a sensitivity analysis to characterise the effects of uncertainties in Ga on the decoupling coefficient.*

*I think the authors have done a thorough job in addressing my comments and I did not find any further issues. I have only a few more minor comments that the authors might want to take on board:*

**[Response]** We thank the Reviewer for the positive comments and helpful suggestions. Please find below the Reviewer's comments (italics), followed by our responses (roman font), with red color indicating relevant changes in the manuscript.

*- I suggest to report measurement height somewhere (maybe Table S1)*

**[Response]** The measurement height has been added to Table S1.

*L. 614: I understand this point better now. Maybe rephrase to "under the whole range of conditions" or similar. But still wondering if 'no wind' ever occurs, maybe rephrase to 'very low wind speeds' or similar.*

**[Response]** Thanks for these suggestions. We have revised accordingly.

*L. 625: should this reference be Thom et al. 1972 (not 1975)?*

**[Response]**. The original reference is Thom et al. 1972, we revised it in the final manuscript.

*L. 627: et al.*

**[Response]** Corrected.

*L. 648: that is interesting, and thanks for the clarification. It would be good to give a % range of what 'significant biases' mean*

**[Response]** The reference paper did not provide the percentage but only the biases in W $m^2$, which we have added to the manuscript. L. 417: "A recent study (McColl, 2020) showed that the linear approximation of Clausius-Clapeyron relation in the Penman-Monteith equation can contribute ~5.7 W $m^2$ biases to daytime and ~1.2 W $m^2$ biases

to nighttime LE. This bias is remarkable when there is large difference between ambient air temperature and surface temperature (often with small Ga)."